# Role of Peroral Cholangioscopy in the Diagnosis of Primary Sclerosing Cholangitis

**DOI:** 10.3390/diagnostics10050268

**Published:** 2020-04-29

**Authors:** Toshio Fujisawa, Mako Ushio, Sho Takahashi, Wataru Yamagata, Yusuke Takasaki, Akinori Suzuki, Yoshihiro Okawa, Kazushige Ochiai, Ko Tomishima, Shigeto Ishii, Hiroaki Saito, Hiroyuki Isayama

**Affiliations:** Department of Gastroenterology, Juntendo University Graduate School of Medicine, 113-8421 Tokyo, Japan; t-fujisawa@juntendo.ac.jp (T.F.); m-ushio@juntendo.ac.jp (M.U.); sho-takahashi@juntendo.ac.jp (S.T.); w.yamagata.mx@juntendo.ac.jp (W.Y.); ytakasa@juntendo.ac.jp (Y.T.); suzukia@juntendo.ac.jp (A.S.); y.okawa.kl@juntendo.ac.jp (Y.O.); k.ochiai.qd@juntendo.ac.jp (K.O.); tomishim@juntendo.ac.jp (K.T.); sishii@juntendo.ac.jp (S.I.); hiloaki@juntendo.ac.jp (H.S.)

**Keywords:** primary sclerosing cholangitis, cholangioscopy, IgG4-related sclerosing cholangitis, cholangiocarcinoma, scarring, pseudodiverticula, active phase, chronic phase

## Abstract

Primary sclerosing cholangitis (PSC) is characterized by idiopathic biliary stricture followed by progressive cholestasis and fibrosis. When diagnosing PSC, its differentiation from other types of sclerosing cholangitis and cholangiocarcinoma is necessary. The cholangioscopic findings of PSC have not been investigated sufficiently. PSC and IgG4-related sclerosing cholangitis are difficult to distinguish by peroral cholangioscopy (POCS), but POCS is useful for excluding cholangiocarcinoma. POCS findings vary according to the condition and stage of disease. In the active phase, findings such as mucosal erythema, ulceration, fibrinous white exudate, and an irregular surface are observed and may reflect strong inflammation in the biliary epithelium. On the other hand, findings such as scarring, pseudodiverticula, and bile duct stenosis appear in the chronic phase and may reflect fibrosis and stenosis resulting from repeated inflammation. Observation of inside the bile duct by POCS might confirm the current PSC activity. Because POCS offers not only information regarding the diagnosis of PSC and PSC-associated cholangiocarcinoma but also the current statuses of biliary inflammation and stenosis, POCS could significantly contribute to the diagnosis and treatment of PSC once the characteristic findings of PSC are confirmed by future studies.

## 1. Introduction

Primary sclerosing cholangitis (PSC) is a chronic cholestatic disease characterized by multifocal and progressive idiopathic strictures of the biliary tree [1]. Progression of chronic cholestasis can lead to liver cirrhosis and ultimately liver transplantation [2,3]. For diagnosis of PSC, radiological examinations are the most useful [4]. Endoscopic retrograde cholangiography (ERC) is the standard diagnostic technique [5]; however, recent improvements in the resolution of magnetic resonance cholangiography (MRC) have resulted in its use for the initial screening and prognostic determination of PSC, with acceptable sensitivity and specificity [6,7,8,9]. Furthermore, MRC is reported to be more cost-effective than ERC [10,11,12]. For PSC diagnosis, after excluding secondary sclerosing cholangitis (SC), differentiation from IgG4-related SC (IgG4-SC) and cholangiocarcinoma is necessary [13]. IgG4-SC is characterized by a relatively good prognosis and good response to steroids [14]. The clinical characterizations of PSC and IgG4-SC, as well as their treatments, are different [15]. Therefore, the differentiation between both of them is important [16].

Dominant stricture, defined as stenosis ≤ 1.5 mm in the common bile duct and/or ≤ 1 mm in the hepatic duct within 2 cm of the hilum, is frequently observed in PSC patients [17] and causes jaundice and cholestatic liver dysfunction (Figure 1A). Dominant stricture needs to be distinguished from cholangiocarcinoma [18,19]. The increased risk of biliary cancer in PSC patients has been firmly established. In a multicenter study, hepatobiliary cancer was diagnosed in 10.9% of the 7119 PSC patients [20]. Moreover, up to 50% of cholangiocarcinomas are detected within a year of the initial PSC diagnosis, and the yearly incidence is estimated to be 0.5%–1.5% [21,22]. Therefore, excluding biliary cancer is very important for continuing treatment of PSC.

## 2. Radiological Images of PSC

PSC has characteristic cholangiogram findings [4], including a band-like stricture, which is relatively short (1–2 mm) compared with the IgG4-SC stricture and exists multifocally in the biliary tree (Figure 1A). The following findings are also observed in PSC: (1) A pruned-tree appearance, which occurs due to the narrowing of the intrahepatic duct, in early-stage disease and decreased arborization of the intrahepatic duct and pruning in advanced-stage disease; (2) a beaded appearance that results from repeating short, annular strictures alternating with normal or minimally dilated segments in the intrahepatic duct (Figure 1B); (3) diverticulum-like outpouching resembling diverticula that protrude between adjacent strictures (Figure 1C) (this finding is often observed in relatively thicker ducts including hepatic ducts and common bile ducts); and (4) a shaggy appearance due to mural irregularities, which produce a characteristic uneven duct wall without strictures. When these findings are observed simultaneously, a diagnosis of PSC is strongly considered. However, there are no specifications regarding the number or grade of these findings indicating a diagnosis of PSC. Therefore, individual physicians are responsible for determining whether the patient has findings characteristic of PSC. 

## 3. Excluding Other Causes of Biliary Stricture 

A Japanese research team published a clinical guideline for diagnosing PSC [23]. This guideline includes a diagnostic flow chart for cases of suspected PSC with chronic cholestasis and/or bile duct dilation. Based on the flow chart, secondary SC should be excluded first. Causes of secondary SC include (1) congenital SC, (2) chronic obstruction, (3) infection, (4) toxicity, (5) immunologic dysfunction, (6) ischemia, and (7) infiltrative disorders [24] (Table 1). In some cases, a medical examination, blood test, and liver biopsy may help exclude these causes of secondary SC [25]. 

After excluding secondary SC, IgG4-SC should be excluded. Cholangiography reveals repeated short-distance strictures in patients with PSC versus relatively long, segmental strictures in patients with IgG4-SC [26]. Moreover, stricture in the intrapancreatic bile duct is common in IgG4-SC, but not in PSC [13] (Figure 2). Table 2 shows a comparison of the characteristic findings between PSC and IgG4-SC.

Computed tomography and cholangiography are the standard techniques for detecting cholangiocarcinoma during the follow-up of PSC; however, their sensitivities (82% and 80%, respectively) and specificities (54% and 53%, respectively) are too low [27]. Njei et al. compared the cost-effectiveness for diagnosis of cholangiocarcinoma associated with PSC among brush cytology only, cytology plus fluorescence in situ hybridization, biopsy sampling under fluoroscopy, and target biopsy sampling with single-operator peroral cholangioscopy (POCS) [28]. The results showed that target biopsy sampling with single-operator POCS was the most cost-effective.

## 4. POCS for Diagnosing PSC

The cholangiographic characteristics of MRC and ERC, but not POCS, have been well investigated. Tischendorf et al. evaluated the usefulness of POCS for detecting cholangiocarcinoma in PSC patients [29]. They prospectively examined 53 PSC patients with dominant strictures and detected cholangiocarcinoma in 12 (23%) of these patients. They showed that POCS was significantly superior to ERC for detecting malignancy at the site of the dominant stricture. The sensitivity, specificity, accuracy, positive predictive value, and negative predictive value of POCS were 92%, 93%, 93%, 79%, and 97%, respectively, all of which were sufficiently high. On the other hand, Arnero et al. prospectively examined the utility of POCS-guided sampling for detection of cholangiocarcinoma in 64 strictures in 47 PSC patients and detected only 3 (4%) malignant lesions [30]. The sensitivity, specificity, accuracy, and negative predictive value of POCS-guided sampling were 33%, 100%, 96%, and 95%, respectively [30]. The sensitivity was much lower than that obtained by Tischendorf et al. The authors suggested that the low sensitivity might have been attributed to the low incidence of cholangiocarcinoma among the subjects. Kalaitzakis et al. examined 52 SC patients, comprising 48 with PSC and four with IgG4-SC, and reported the effectiveness of POCS for diagnosing cholangiocarcinoma (50% sensitivity and 100% specificity) [31] (Table 3). The sensitivity of POCS might not be high enough to detect cholangiocarcinoma in primary SC.

## 5. The Phases of PSC and Cholangioscopic Findings in Each Phase

In terms of differentiation by appearance, Itoi et al. examined the usefulness of cholangiography for differentiating PSC, IgG4-SC, and cholangiocarcinoma [32]. They classified cholangioscopic findings into the following nine features: (1) mass formation, (2) bile duct stenosis, (3) dilated vessels, (4) tortuous vessels, (5) partially enlarged vessels, (6) irregular surface, (7) scarring, (8) pseudodiverticula, and (9) friability. As a result, scarring and pseudodiverticula were found to be characteristic findings of PSC, and partially enlarged vessels were characteristic of cholangiocarcinoma. The other findings of mass formation, dilated vessels, tortuous vessels, irregular surface, and friability were observed in both IgG4-SC and cholangiocarcinoma and were not helpful for distinguishing the two diseases. An irregular surface, although rare, was also observed in PSC. Sandha et al. further investigated the cholangioscopic findings of PSC and stratified PSC into the following three types based on visual characteristics: (1) inflammatory, (2) fibrostenotic, and (3) nodular or mass-forming. The inflammatory type showed mucosal erythema, ulceration, and fibrinous white exudate; the fibrostenotic type showed circumferential rings and asymmetric cicatrization; and the nodular or mass-forming type showed focal nodular tissue growth [33]. Although the small numbers of patients in both studies (five [32] and 29 [33] patients) might not accurately represent the overall disease characteristics of PSC, these results suggest that the cholangioscopic findings of PSC may vary according to the disease stage. Itoi et al. proposed that scarring and pseudodiverticula might only be indicative of the fibrostenotic type of PSC. The presence of the nodular or mass-forming type should signal the possibility of cholangiocarcinoma. We also experienced cholangiography in our PSC patients, and all of the findings, except partially enlarged vessels, were observed. Mucosal erythema (Figure 3A), ulceration (Figure 3B), fibrinous white exudate (Figure 3C), and irregular surface (Figure 3D) are often observed in active-phase patients with repeat acute cholangitis. These findings appear during the relatively early stages of PSC and might be classified as the inflammatory type, as proposed by Sandha et al. Patients with this type of PSC are also likely to have dilated vessels (Figure 4A), tortuous vessels (Figure 4B), friability (Figure 4C), and mass formation (Figure 4D). However, these findings are commonly observed in cholangiocarcinoma cases. On the other hand, scarring (Figure 5A), pseudodiverticula (Figure 5B), and bile duct stenosis (Figure 5C) are often observed in chronic-phase patients with a long history of PSC. Such patients might be classified as the fibrostenotic type. The finding of mass formation is sometimes observed in both the active and chronic phases of PSC. Since mass formation is difficult to distinguish from cholangiocarcinoma, it should be noted regardless of the phase. The time course of each phase and the characteristics of the different PSC types according to phase are summarized in Figure 6.

## 6. Diagnosis and Classification of PSC Using POCS

Due to advancements in endoscopy, POCS is used not only for clinical research but also as a part of daily medical practice [34,35]. It is often used to diagnose cancer and treat stones, and extensive knowledge and experience have accumulated in those fields [36,37,38]. As mentioned above, however, there are few studies on the use of POCS for PSC and thus insufficient evidence to form a consensus on the diagnosis and treatment of PSC [39,40]. At present, MRCand ERCfindings of the bile duct are important for the diagnosis of PSC, but the results of these techniques strongly reflect the skill and subjectivity of the physician who made the diagnosis [4]. Although POCS is still useful for the diagnosis of cholangiocarcinoma in PSC, POCS will likely make a more significant contribution to the definitive diagnosis of PSC once the characteristic findings of PSC are confirmed in future studies. In addition, POCS may be useful for predicting the time to liver transplantation and classifying the disease stage, which will allow the pharmacotherapeutic effects to be evaluated. Since POCS has great potential for PSC treatment, further examination and accumulation of knowledge is necessary.

## Figures and Tables

**Figure 1 diagnostics-10-00268-f001:**
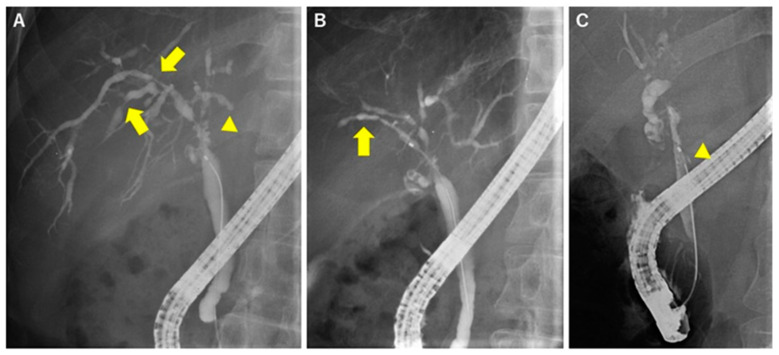
Representative cholangiographic findings of primary sclerosing cholangitis (PSC). (**A**) Dominant stricture (arrowhead) and multiple band-like strictures (arrow). Cholangiogram shows a short (1.5 cm) stricture at the hilum and a few short strictures at the intrahepatic bile ducts. (**B**) Beaded appearance (arrow) and (**C**) diverticulum-like outpouching (arrowhead).

**Figure 2 diagnostics-10-00268-f002:**
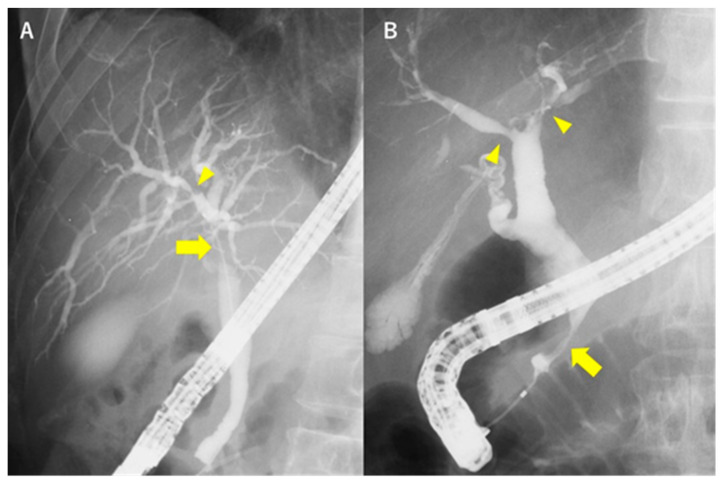
Comparison of the endoscopic retrograde cholangiography (ERC) findings between PSC and IgG4-SC. (**A**) ERC of PSC shows a dominant stricture (arrow) in the hilar part and short (band-like) stricture (arrowhead) at the intrahepatic bile duct. (**B**) ERC of IgG4-SC shows relatively long strictures in the intrahepatic bile duct (arrowhead) along with other strictures at the intrapancreatic bile duct (arrow).

**Figure 3 diagnostics-10-00268-f003:**
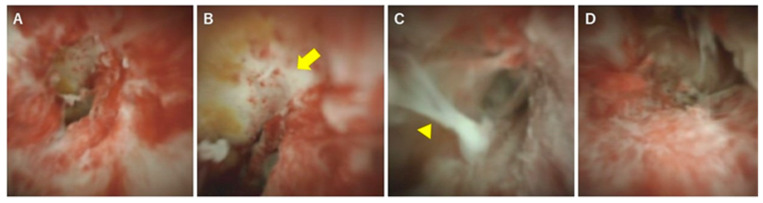
Cholangioscopic findings characteristic of the active phase of PSC. (**A**) Mucosal erythema, (**B**) ulceration (yellow arrow), (**C**) fibrinous white exudate (arrowhead), and (**D**) irregular surface. These findings are observed mainly during the active phase of PSC.

**Figure 4 diagnostics-10-00268-f004:**
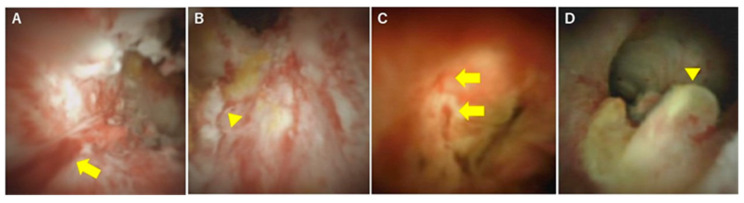
Cholangioscopic findings distinguishing PSC from malignancy. (**A**) Dilated vessels (yellow arrow), (**B**) tortuous vessels (arrowhead), (**C**) friability, oozing with saline irrigation alone (arrows), and (**D**) mass formation (arrowhead).

**Figure 5 diagnostics-10-00268-f005:**
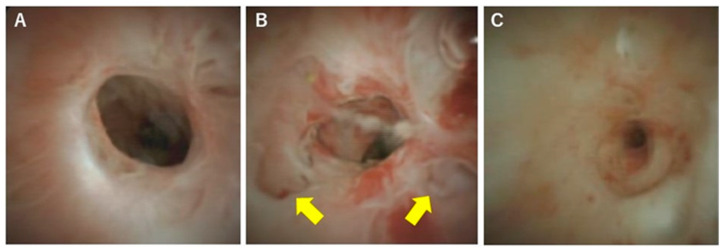
Cholangioscopic findings characteristic of the chronic phase of PSC. (**A**) Scarring, (**B**) pseudodiverticula (yellow arrows), and (**C**) bile duct stenosis. These findings are observed mainly during the chronic phase of PSC.

**Figure 6 diagnostics-10-00268-f006:**
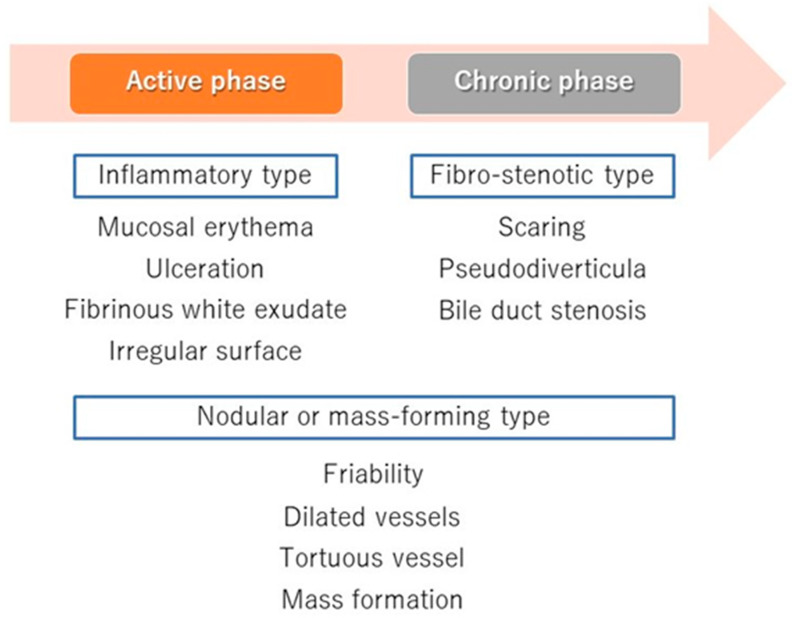
Time course of the PSC phases and the types and cholangioscopic findings of each phase. The active phase represents early-stage PSC and is characterized by the cholangioscopic findings of the inflammatory type (mucosal erythema, ulceration, fibrinous white exudate, and irregular surface). The chronic phase represents late-stage PSC and is characterized by the findings of the fibrostenotic type (scarring, pseudodiverticula, and bile duct stenosis). Dilated vessels, tortuous vessels, friability, and mass formation in the mass-forming type of PSC can occur in either phase.

**Table 1 diagnostics-10-00268-t001:** Causes of secondary sclerosing cholangitis

Causes	Diseases and Pathogens	Diagnosis
Chronic obstruction	Choledocholithiasis	US, MRC/ERC
	Idiophathic biliary strictures	Exclusion diagnosis
	Neoplasms (benigh and malignancy)	US, CT/MRI, ERC with biopsy
Congenital	Caroli’s disease	History, CT/MRI
	Cystic fibrosis	Sweat chloride ion, family history, CFTR * gene
Immunologic	Eosinophilic cholangitis	CT/MRC, ERC with biopsy
	Graft-versus-host disease	History, liver biopsy
Infections	Bacteria (Recurrent pyogenic cholangitis)	History, CT/MRI
	Virus (cytomeralovirus and HIV *)	Serology, immunological investigations
	Parasite	CT/MRI, ERC, serology
Infiltrative disorders	Systemic vasculitis	Biopsy, serum ANCAs *, angiography
	Amyloidosis	Symptoms, liver biopsy
	Sarcoidosis	Serum ACE *, liver biopsy
	Systemic mastocytosis	Bone marrow biopsy
	Heypereosinophilic syndrome	Eosinophilia, bone marrow biopsy
	Hodgkin’s disease	CT/MRI, serum sIL-2R *, histology
Ischemic	Vascular trauma	History, CT/MRI
	Anastomotic strictures in liver graft	History, CT/MRI
	Transcatheter arterial embolization	History
Toxic	Alcohol, formaldehyde, hypertonic saline	History
	Intraarterial chemotherapy	History
Trauma	Accident, Surgery, and ERCP	History, CT/MRI
	Radiation injury	History

* HIV: human immunodeficiency virus, CFTR: cystic fibrosis transmembrane conductance regulator, ANCAs: antineutrophil cytoplasmic antibodys, ACE: angiotensin-1-converting enzyme, sIL-2R: soluble interleukin-2 receptor

**Table 2 diagnostics-10-00268-t002:** Comparison of the characteristic findings between PSC and IgG4-related SC (IgG4-SC).

Favors PSC	Characteristic	Favors IgG4-SC
Short, multiple	Length of stricture	Long
Rare	Stricture of intrapancreatic bile duct	Often
Pruning	Intrahepatic bile ducts	Dilating

**Table 3 diagnostics-10-00268-t003:** Summary of the cholangiocarcinoma diagnostic ability of peroral cholangioscopy (POCS) in PSC patients.

Author	Year	Number of Patients	Sensitivity	Specificity	Accuracy	PPV	NPV
Tischendorf et al.	2006	53	92%	93%	93%	79%	97%
Kalaitzakis et al.	2014	52 (including 4 IgG4-SC)	50%	100%	88%	100%	87%
Arnelo et al.	2015	47	33%	100%	96%	100%	95%

PPV: positive predictive value, NPV: negative predictive value.

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
