# Peer review of "Role of Peroral Cholangioscopy in the Diagnosis of Primary Sclerosing Cholangitis"

_diagnostics, 2020, doi:10.3390/diagnostics10050268_

Round 1

Reviewer 1 Report

Dear Authors, 

I would like to congratulate you on your choice of work topic.

Below you can find major and minor comments.

Major

  1. Fluoroscopic evaluation during ERCP procedure is the most important step for further management, as was mentioned. Therefore, in my opinion, for better visualization and differentiation PSC vs IgG4-SC, I would like to add the table in section 2. (PSC vs IgG4 radiological differentiation). Also, photos would be appreciated, to showed the difference between both types of this diseases with appropriate marking (maybe with different colors).
  2. A meta-analysis of 6 studies [*] compared MRCP to ERCP for the diagnosis of PSC found MRCP to have a high sensitivity and specificity. Also, cost-minimization analyses have found a significant cost savings by performing MRCP, what should be mentioned {**}. For visualization of the different strictures Fig.3 in ** could be add.
  3. Data from Tischendorf study (line 93) are too poor to convince the reader about the effectiveness of this method. The data should be supplemented with for e.g. the number of patients, or the fact that it was the study with the largest group of cases assessed so far. Similarly, in the study of Arnero (line 97).
  4. Please consider adding a table that compares current study results POCS for PCS with all listed parameters in lines 95, 96 etc. It is not entirely clear.
  5. Why the sensitivity in Arnero’s study is lower?
  6. The text from 103 to 134 should be clearly presented. Please consider to use a separate paragraph to describe the individual features of these three types of pathologies for individual authors- Itoi and Sandha, maybe adding a table or diagrams that will order important features visible in POCS.

*Dave M, Elmunzer BJ, Dwamena BA, Higgins PD. Primary sclerosing cholangitis: meta-analysis of diagnostic performance of MR cholangiopancreatography. Radiology. 2010;256:387–396.

** Fung BM, Tabibian JH. Biliary endoscopy in the management of primary sclerosing cholangitis and its complications. Liver Res. 2019;3(2):106–117. doi:10.1016/j.livres.2019.03.004 –

Minor

Please change the sentence in lines:

12-abbreviation SCc not explained (….)

17-on the other hand

32,33- Traditionally…. (not correct)

75- please to fit the table on one page, now is separated

91- Characteristi…..language in this sentence should be changed

107- characteristic rather than characteristical

104 and 105- Here we have features differentiated three types of changes. Please extract the characteristic features for eg. in a table, what may be helpful for the differentiation these three types pathologies. This is a very important conclusion, however now it is illegible in this part of manuscript.

Author Response

We appreciate the reviewer for the valuable suggestions. We made a point-by-point response to the comments

Major

  1. Fluoroscopic evaluation during ERCP procedure is the most important step for further management, as was mentioned. Therefore, in my opinion, for better visualization and differentiation PSC vs IgG4-SC, I would like to add the table in section 2. (PSC vs IgG4 radiological differentiation). Also, photos would be appreciated, to showed the difference between both types of this diseases with appropriate marking (maybe with different colors).

→We appreciate the reviewer for the valuable suggestions. We added the table and figure for comparing PSC and IgG4 in the section “Excluding other causes of biliary stricture”.

  1. A meta-analysis of 6 studies [*] compared MRCP to ERCP for the diagnosis of PSC found MRCP to have a high sensitivity and specificity. Also, cost-minimization analyses have found a significant cost savings by performing MRCP, what should be mentioned {**}. For visualization of the different strictures Fig.3 in ** could be add.

→* Dave’s paper has been already included in the submitted version. ** Fung’s paper was newly added with the explanation of cost effectiveness.

  1. Data from Tischendorf study (line 93) are too poor to convince the reader about the effectiveness of this method. The data should be supplemented with for e.g. the number of patients, or the fact that it was the study with the largest group of cases assessed so far. Similarly, in the study of Arnero (line 97).

→Thank you for your helpful suggestion. We added the number of patients in the manuscript and made the new table (Table 3) showing the essences of 3 presented papers, which examined the usefulness of SOCS for detecting cholangiocarcinoma in the PSC patients.

  1. Please consider adding a table that compares current study results POCS for PCS with all listed parameters in lines 95, 96 etc. It is not entirely clear.

→We made the new table (Table 3) showing the essences of 3 presented papers, which examined the usefulness of SOCS for detecting cholangiocarcinoma in the PSC patients.

  1. Why the sensitivity in Arnero’s study is lower?

→The authors themselves considered that low sensitivity could be due to the low incidence of cholangiocarcinoma in the subjects. Therefore, we added the sentence in the manuscript. In fact, the sensitivity of POCS might be not enough high for detecting cholangiocarcinoma hiding in primary sclerosing cholangitis

  1. The text from 103 to 134 should be clearly presented. Please consider to use a separate paragraph to describe the individual features of these three types of pathologies for individual authors- Itoi and Sandha, maybe adding a table or diagrams that will order important features visible in POCS.

→We made a new paragraph that explaining the PSC phase and the cholangioscopic findings in each type. And we made a new figure (figure 6) presenting the time course of PSC phase and the findings.

*Dave M, Elmunzer BJ, Dwamena BA, Higgins PD. Primary sclerosing cholangitis: meta-analysis of diagnostic performance of MR cholangiopancreatography. Radiology. 2010;256:387–396.

** Fung BM, Tabibian JH. Biliary endoscopy in the management of primary sclerosing cholangitis and its complications. Liver Res. 2019;3(2):106–117. doi:10.1016/j.livres.2019.03.004 –

Minor

Please change the sentence in lines:

12-abbreviation SCc not explained (….)

→We spelled it out “sclerosing cholangitis”.

17-on the other hand

→Corrected.

32,33- Traditionally…. (not correct)

→It was removed.

75- please to fit the table on one page, now is separated

→The table was made small, and fit in the page.

91- Characteristi…..language in this sentence should be changed

→We corrected the sentence.

107- characteristic rather than characteristical

→“Characteristical” was changed into “characteristic”.

104 and 105- Here we have features differentiated three types of changes. Please extract the characteristic features for eg. in a table, what may be helpful for the differentiation these three types pathologies. This is a very important conclusion, however now it is illegible in this part of manuscript.

→There is no report that examines the relationship the finding of SOCS and pathologies in PSC. We arranged the findings and type into the time course and the phase of PSC and summarize them into the figure 6. It may help the readers understand the relationship between the finding, types and phases.

Reviewer 2 Report

The study is aimed to review Peroral Cholangioscopy for diagnosis of primary sclerosing cholangitis.        The title is “A Role of Peroral Cholangioscopy for Diagnosis of Primary Sclerosing Cholangitis”.

  1. This is a review article.  
  2. Several factors influence the outcome of this technique. Please discuss these.
  3. Please summarize and compare this technique and the standard method in the “Table”.
  4. Please review the literature and add more details in the discussion section.
  5. What is the new knowledge from this report?
  6. Finally, please recommend the readers “How to apply this knowledge for routine clinical practice?”.

Author Response

We appreciate the reviewer for the valuable suggestions. We made a point-by-point response to the comments

The study is aimed to review Peroral Cholangioscopy for diagnosis of primary sclerosing cholangitis. The title is “A Role of Peroral Cholangioscopy for Diagnosis of Primary Sclerosing Cholangitis”.

  1. This is a review article.  
  2. Several factors influence the outcome of this technique. Please discuss these.

→Thank you for the reviewer’s suggestion. →We made a new paragraph that explaining the PSC phase and the cholangioscopic findings in each type. And we made a new figure (figure 6) presenting the time course of PSC phase and the findings.

  1. Please summarize and compare this technique and the standard method in the “Table”.

→We newly made two tables and two figures, which will help the readers understand the manuscripts.

  1. Please review the literature and add more details in the discussion section.

→We did not made discussion part in this manuscript because of the review article. However, In the last paragraph, we summarize the present situation and shows proposal to the future of PSC.

  1. What is the new knowledge from this report?

→This is the review article, so we summarized the present situation of SOCS for PSC clinic. We believe this review helping the readers their clinical work.

  1. Finally, please recommend the readers “How to apply this knowledge for routine clinical practice?”.

→We summarized the SOCS findings in each PSC type into the figure 6. It helps the readers for diagnosing PSC using SOCS.

Round 2

Reviewer 1 Report

Thank you for considering all comments and suggestions. 

After corrections manuscript is clearly presented, especially with diagrams and tables, what is a great value for not experts in this field. 

Only one and final suggestion is to change one sentence, optionally.

"The clinical characterization of, and treatments for, PSC and IgG4-

SC are completely different15; therefore, differentiation between the two types of SC is important "

Maybe better:

The clinical characterization of PSC and IgG4-SC, as well as treatment, are different. Therefore, the differentiation between both of them is important.

Best regards,

Reviewer

Author Response

We appreciate the reviewer for the valuable suggestions. We changed the sentence of the indicated part following the reviewer’s suggestion. The changes are indicated in red in the text.

Reviewer 2 Report

The revised version is O.K.

Author Response

We appreciate the reviewer for the valuable suggestions.